# Association between Living with Children and the Health and Health Behavior of Women and Men. Are There Differences by Age? Results of the “German Health Update” (GEDA) Study

**DOI:** 10.3390/ijerph17093180

**Published:** 2020-05-02

**Authors:** Petra Rattay, Elena von der Lippe

**Affiliations:** Department of Epidemiology and Health Monitoring, Robert Koch Institute, 12101 Berlin, Germany; E.vonderLippe@RKI.de

**Keywords:** family, living with children, parenthood, mother, father, self-rated health, health behavior, mental health, Germany

## Abstract

Does the health of women and men living with and without minor children differ, and are age differences evident in the association? For self-rated general health, depression, back pain, overweight, smoking and sporting inactivity, the GEDA data 2009–2012 (18–54 years, n = 39,096) were used to calculate prevalence for women and men stratified by parental status (living with children: yes/no) and age. Moreover, we calculated odds ratios and predictive margins, performing logistic regressions with interaction terms of parental status and age. Women and men aged 45–54 living with children are healthier than those not living with children. Parents aged 18–24 smoke more frequently and do less sport; young mothers are also more likely to be overweight and suffer from back pain than women not living with children. In multivariable analysis, the interaction of living with children and age is significant for all outcomes (except depression and back pain in men). Living with children is an important social determinant of health, highly dependent on age. It is to be discussed whether the bio-psycho-social situation has an influence on becoming a parent, or whether parenthood in different phases of life strains or enhances health.

## 1. Introduction

Living together with children can contribute to good health through close social and emotional relationships as well as mechanisms of structuring everyday life, social control and meaningfulness of life. On the other hand, living with children is associated with a number of demands and obligations as well as conflicting role expectations that can lead to stress and poor health [1,2,3]. This is especially true when problems in the partnership or the sole responsibility for the upbringing of the children, financial worries or low social support occur. In addition to the effects of co-residence with children on the health of women and men (“causality” hypothesis), the health and health behavior of women and men can also have an influence on the probability of finding a partner or starting a family (“selectivity” hypothesis) [2,4].

### 1.1. State of Research

The current state of research on the health and health behavior of parents is inconsistent. Whereas some studies have reported better health and healthier behavior among parents than among women and men without children, other research reports poorer health among parents. In turn, some studies have found no association between parenthood and health. 

Firstly, this variation is related to the large number of different health outcomes investigated. Inconsistent results have been found in particular in self-rated general health [5,6,7,8,9,10,11,12,13] and mental health [1,13,14,15,16,17,18,19,20,21,22,23,24,25]. For overweight or obesity, however, the majority of studies have reported higher prevalence or a higher BMI for mothers [13,26,27,28,29,30,31], while the findings for men are heterogeneous [13,32]. For physical activity and in particular for the exercise of sports, results are also found to the disadvantage of parents [27,33,34,35,36,37,38]. However, more household-related physical activity has been evident among women and men with children [39]. The diet of mothers follows the current recommendations to a greater extent than of nonparents [40]; nevertheless, mothers consume more calories [27,41]. With regard to tobacco and alcohol consumption, women and men with children behave more healthily than those without children [42,43,44,45,46,47,48]. 

Besides the dependence on health outcome, results on the association between parenthood and health also vary with the number and age of children [26,40,49,50,51] as well as the partner, employment and socioeconomic statuses [52,53]. For example, many studies have reported a higher psychosocial and health burden especially for single parents [5,15,16,54,55,56]. There are also differences by gender in the association between parental status and health, with significantly more studies analyzing the health of mothers (e.g., [7,8,9,10,12,17,23,55,57]) than of fathers [31,36,58,59,60,61,62,63]. Gender-comparative studies, in turn, do not produce consistent results (e.g., [54,64,65,66,67,68,69]), so that it is not possible to assess in general terms whether the health of parents is associated with gender. However, the social system and welfare state regulations also seem to influence the association between parental status and health. International comparisons have supported the thesis that parenthood in the USA is associated to a greater extent with poor health than in Western and especially Northern European countries, for which positive links between parenthood and health have often be found [44,64,70,71,72,73,74]. Hansen [75] assumes that these country-specific differences can be attributed to the fact that the Nordic welfare states offer families much more comprehensive public support (such as available and affordable daycare, flexible work schedules, job leave security, cash benefits, and paid parental leave) than the USA or Australia.

Besides this, it can be assumed that a large part of the inconsistent results can be attributed to their relationship to different age groups [76]. Although most results are adjusted for age, the moderating effect of age on the association between parenthood and health is rarely analyzed explicitly. When age-stratified results are reported, almost all studies have shown that very young parents in particular are severely affected by health problems [5,55,76,77,78,79,80]. This is supported by studies that include the age at first birth. The birth of the first child shortly after and especially before the mother’s 20th birthday is associated with poorer health [12,20,25,52,78,81], less healthy behavior [45] and higher mortality [61,82].

Some studies have found that women and men who became parents late in life have a lower mortality rate and higher life satisfaction than childless women and men [61,73,82,83,84]. For example, in Margolis and Myrskylä [79] the association between parenthood and happiness at the age of 15 to 29 is negative, while it is not significant at the age of 30 to 39 and positive from the age of 40. Other studies have suggested that—especially in women—an age of around 30 years at the birth of the first child seems to be associated with good maternal health [20,85,86]. According to Graham [57] mothers show better physical and mental health and well-being compared to childless women, especially at the age of 34 to 44; while childless women perform better at a very young age and in old age. Carlson [85] has found that a deviation from the anticipated age at birth of the first child in younger and older mothers is accompanied by an increase in depressive symptoms.

With regard to Germany, Helbig et al. [18] have concluded that for 18- to 49-year-olds there is no moderating effect of age on the association between parental status and mental health. Stöbel-Richter et al. [23] have also found no age differences in the association of parenthood with depression or anxiety in 18- to 50-year-old women. In an analysis by a German health insurance company [78] on sick days of employed women and men with and without family-insured children, there are no differences in parental status in the middle working age (30–44 years), while young working parents have a slightly above average number of sick days and older men with children have a below average number of sick days. Becchetti et al. [87] have found higher happiness among parents compared to nonparents only from the age of 55 onwards. In a longitudinal study with data from the German Socio-Economic Panel, Myrskylä and Margolis [83] have shown that in women and men who became parents late in life (from the age of 35), happiness increased with the birth of the first and second child and remained then at a stable level for a longer period of time. On the other hand, happiness decreased in women and men who became parents between the ages of 18 and 22. Women and men who became parents at ages 23 to 34 have increasing happiness before a first birth and in the year of birth, but afterwards happiness decreases to the baseline level or below.

The short overview of the international and national research on the health of parents and on the moderating effect of age on the association between parenthood and health shows: (a) that the association between the parental status and different health outcome parameters varies, (b) that international findings on the health of parents cannot easily be transferred to Germany due to the varying social systems and family policies, and (c) that so far there are only a few gender-comparative studies on age differences regarding the association between parental and health status. 

### 1.2. Aim of the Study

The aim of this analysis is to compare the health and health behavior of women and men living with and without children in Germany and to explore whether age differences in this association can be observed. 

The research questions are:Are there differences in the health and health behavior of women and men according to parental status (living with children)?Does the association between health/health behavior and parental status in women and men vary with age?

## 2. Materials and Methods

### 2.1. Data

For the analyses, the pooled data of the study “German Health Update” (GEDA) from 2009, 2010 and 2012 were used, which were carried out by the Robert Koch Institute as part of health monitoring [88]. The GEDA study is a nationally representative telephone survey of German-speaking adults who live in private households and have a landline phone connection. Data were collected using computer-assisted telephone interviews. Random samples of telephone numbers from the German fixed-line network were generated, adapting the Gabler–Häder method [89]. The sample consists of 62,606 women and men aged 18 and over. The cooperation rate at respondent level (the proportion of interviews that were completed after initial contact with a potential participant), was 51.2% in 2009, 55.8% in 2010 and 76.7% in 2012. The response rate (the proportion of completed interviews to the number of neutral non-responses in the adjusted gross sample) amounted to 29.1% in 2009, 28.9% in 2010 and 22.1% in 2012 [88]. The absolute number of cases per survey year in the total sample were 21,262 in 2009, 22,050 in 2010 and 19,294 in 2012.

The GEDA surveys of the years 2009–2012 provide a large representative data set on health and disease, determinants of health, and the use of health services for use in national and European health reporting systems, health policy, and public health research [88]. Owing to the sample size of the pooled data, it is well suited for the analysis of subsamples.

Each GEDA round was approved by The Federal Commissioner for Data Protection and Freedom of Information, and verbal informed consent was obtained in advance from all participants. 

For this analysis, the sample was limited to participants aged 18 to 54. The number of cases of women aged 55 years and above living with minor children is relatively small. After plausibility checks of the age data of the household members, 50 participants were excluded, so that the analyses were based on data from 39,096 women and men (weighted proportions: women: 49.1%, men: 50.9%). The sample description is given in Table 1. 

### 2.2. Variables

Outcome variables: The analysis included self-rated general health, depression, back pain, overweight, smoking, and sporting inactivity as outcome variables. Self-rated general health was surveyed using the question “How is your general state of health?”. The response categories were dichotomized into “very good/good” versus “fair/poor/very poor”. Depression was assessed by asking whether the respondent was suffering from depression or depressive mood in the last 12 months diagnosed by a doctor or psychotherapist (12-month prevalence). The 12-month prevalence of back pain is based on the question as to whether the respondent has had at least three months of persistent back pain in the last 12 months. Overweight data are based on respondents’ height and weight data. According to the WHO definition, overweight occurs when the Body Mass Index (BMI) is greater than 25 kg/m^2^. Information on smoking (“yes”/“no”) was asked with the question “Do you smoke from time to time—even if only occasionally?” In the present analysis, the response categories “daily” and “occasional” were combined to “yes”. The data on sporting inactivity (“yes”/“no”) are based on the respondents’ self-declaration that they have not practiced any sport in the last 3 months. 

Predictor variable: Parental status is based on information provided by participants on all persons living in the household (relationship to respondent and age). We define women and men as parents when they live together with at least one own child under 18 years. We did not differentiate among the respondents’ biological children, adopted children, or stepchildren (social parenthood). 

Moderator variables: Age and sex are defined as moderator variables. The results of the descriptive analyses are stratified by age groups (18–24 years, 25–34 years, 35–44 years and 45–54 years) while in the multivariable analyses age (in completed life years) is included as a metric variable. All analyses are stratified by sex.

Control variables: Socioeconomic, employment and partner statuses as well as the age of the youngest child and the residential region are taken into account as control variables. The socioeconomic status was determined by means of an index which included information on school and vocational education, occupational status and net household income and which allows classification into “low”, “medium” and “high” status groups [90]. For measurement of employment status (self-defined), we differentiate among “employed full-time”, “employed part-time,” and “not employed.” The partner status takes into account living with a partner in the same household (“yes”/“no”), regardless of the marital status. With regard to the age of the youngest child, living with a pre-school aged child (0–6 years) (“yes”/“no”) was included. Furthermore, we controlled the logistic models for the residential region (“West Germany”/“East Germany incl. Berlin”). 

### 2.3. Data Analysis

In the first step, prevalences were calculated for all outcome variables for women and men with and without children in the household (parental status) stratified by age groups and sex. 

In the second step, we calculated multivariable logistic regressions over the entire age range for each outcome variable. In these models, age was used as a metric variable (completed life years, centered on the minimum age of 18 years). Age modelling (age vs. age squared) was performed by calculating the BIC (Schwarz’s Bayesian information criteria) for selecting the (fully adjusted) model with the best “fit” (=lowest BIC value). For the analysis of the moderating effect of age on the association between parental status and health, the interaction between parental status and age or age squared was included in the models. Depending on the model with the best “fit”, there were three variants of the interaction term: (a) parental status#age, (b) parental status#age and parental status#age squared, (c) parental status #age squared. 

All models were adjusted for socioeconomic, employment and partner statuses, the age of the youngest child and the residential region. Odds ratios and predictive margins (adjusted at the mean) were calculated for all models [91]. In this paper, we present the predictive margins graphically. All calculations were carried out using a weighting factor that corrects for deviations within the sample from the population structure (as of 31 December, 2010) regarding age, sex, education and federal state. The analyses were conducted with the StataSE 15 statistical software (StataCorp, College Station, TX, USA) using the survey (svy) module. Statistical significance in the descriptive as well as in the multivariable analysis was determined using *p*-values (*p* < 0.05).

## 3. Results

Looking at the raw prevalence for the total group of 18- to 54-year-olds (Table 2, first column), it can be seen that women and men living with children are more often overweight and less frequently active in sports than women and men in childless households. Men with children also suffer more frequently from back pain. On the other hand, women and men with children smoke less often than those without children. Moreover, women living in households with underage children are less likely to report poor general health and suffer from depression than women in childless households.

However, a comparison of age-stratified prevalence for women and men with and without children shows that there are significant age differences (Table 2). In the youngest age group (18–24 years), the prevalences for all outcome variables are higher in women and men with children than in women and men without children—although not all differences reach statistical significance. In women, all prevalences (except for those for poor general health) are significantly higher in those living with children. Among men, significant differences according to parental status are only found in back pain, smoking and sporting inactivity. 

In the group aged 45–54, the opposite association can be observed. In this age group, the prevalences of all health outcomes are higher in women and men from childless households than in women and men living with children. In women, for all health outcomes significant differences according to parental status were found, whereas in men the differences in prevalence for depression and overweight are not statistically significant.

The results of the multivariable models including the entire age range can be found in Table 3 and Table 4 (odds ratios and *p*-values). In Figure 1, the predictive margins for each health outcome presenting the interaction of parental status and age are shown separately for women and men. 

For women, the interactions between age and parental status are highly significant for all health outcomes (*p* < 0.001). Figure 1 shows that the association between parental status and health varies greatly with age. Young women (up to approx. 30 years of age) without children are healthier and behave healthier when comparing with women with children of the same age. Among women aged approximately 30–40 years, there are hardly any health differences according to the parental status. For women over the age of 40, those who live with children in the household are healthier and behave more healthily than women without children. From the age of 36, women with children are less likely than women without children to be depressed. It can also be observed that the predicted probabilities in women with children hardly change with age, while significantly higher predicted probabilities are found among women in childless households in the older age groups (except smoking).

The patterns of smoking and sporting inactivity in men are similar to that of women. For self-rated health and overweight, the differences between men with and without children are smaller than for women, but the interaction terms are still significant. For depression and back pain, however, there are no significant interactions between parental status and age in men. 

## 4. Discussion

This paper examines whether there are associations between living with underage children and health as well as health behavior and whether these associations vary with age. 

Without a differentiation according to age, a rather inconsistent picture emerges with regard to the association between parental status and health. For each health outcome, prevalence not stratified for age, nevertheless, largely coincides with the results of international research: women and men living with children are more often inactive in sports and more often overweight than women and men without children [26,27,28,31,33,37]. In contrast, women and men living with children smoke less often than childless women and men [45,48]. Moreover, mothers in particular often rated their general health better than women in childless households [5,6,9,10]. Mothers also suffer less from depression than women living without children, while men show no association between parental status and depression. This gender-differentiated pattern is also found for depression in Nomaguchi and Milkie [1] and Kalucza et al. [19], but not in Helbig et al. [18]. 

When age is taken into account, however, the present analysis reveals a rather clear pattern: the association between parental status and health or health behavior varies greatly with age. This is particularly true for women, where younger mothers are less healthy or behave less healthily and older mothers are healthier or behave more healthily than women of the same age without children. In the age between the end of 20 and the beginning of 40, however, there is only little or no difference in health by parental status. In men, this pattern is similarly pronounced only in health behavior (smoking and sporting inactivity) and slightly less pronounced in self-rated general health and overweight, but not in depression and back pain. This age gradient is consistent with some existing study results, which have also found poorer health for parents in the younger age group and better health or well-being for parents in the older age group than for nonparents of the same age [73,76,78,79,84]. The differences by sex described here are also found in the study by Margolis and Myrskylä [79] in the way that the association between parental status and happiness is greater among women aged 40 and over than among men. In contrast, evaluations of the insured-person data of the Techniker Krankenkasse [78] have shown the age effect particularly among men. However, this could be related to the fact that parenthood is represented in the insured-person data by the indicator of family-insured children, and this indicator identifies women with children worse than men with children. The result that the age differences in the association between co-residence with children and health in our analysis are more pronounced among women than among men could possibly be explained by the fact that the everyday life of women is influenced to a greater extent by living with children than that of men. This is reflected, for example, in greater responsibility for the care and upbringing of children as well as in greater problems in reconciling family and work life [79,92]. All in all, when comparing our findings with other study results, it must be taken into account that parenthood can be defined in different ways (fertility history versus living with children).

It is to be discussed whether the results reported here are to be interpreted as age effects or cohort effects. The observed age effect in the association between parental status and health could be explained on the one hand by the fact that the effect of parental caregiving changes with aging or varies between different phases of life. On the other hand, the described differences by age could also indicate cohort (or period) effects. Margolis and Myrskylä [79] can show, for the association between parenthood and happiness in a comparison of two cohorts, that the age gradient can be measured in both cohorts and thus cannot be attributed to cohort effects.

However, it can be assumed that it is less the age itself than the phase of life and the respective social context that have an influence on whether parental caregiving strains or enhances health. It can be presumed that, especially for young mothers and fathers, the close temporal link between education or entry into the labor market and the social role of parenthood can lead to reconciliation problems that can be accompanied by poorer health. Studies have shown, for example, that young mothers are often still in education at the time of their first child’s birth, have not yet established themselves professionally and live in less stable partnerships, which often leads to social disadvantages in the further course of their lives [80,93]. Although Arnett [94] describes “emerging adulthood” as a “roleless role” in which young women and men have a wide range of opportunities and are much less restricted by role obligations than in all other phases of life, this does not apply equally to men and especially to women, who are already parents in this phase of life. In contrast to partnership, cohabitation, education or occupation, parenthood is not reversible. However, it is also quite conceivable that young women in particular, who have problems finding a satisfactory education and employment (possibly also due to poor health), consciously opt for the early formation of a family and a more family-oriented life plan [95]. Since studies also show that women who became mothers at an early age have experienced strong social disadvantages in childhood [93,96,97,98], it can be assumed that this is also reflected in poor health (social selection processes). 

The analysis has some limitations. A main limitation is that the present analysis was carried out with cross-sectional data that do not allow statements regarding whether parental caregiving influences health or whether the health status influences the probability of starting a family (reverse causality or selection processes). In addition, the results can be distorted by unobserved heterogeneity in that—in addition to the control variables included here—other unobserved personality traits and social factors influence both the health status and the probability of parenthood, but in fact there is no causal relationship between parenthood and health [75,99]. In a review on parenthood and happiness, Hansen [75] concludes that findings from cross-sectional studies on the association between parental status and well-being can only be attributed to a small extent to selection processes and unobserved personality traits. Kalucza et al. [19] also came to the conclusion in longitudinal analyses regarding the association between parenthood and mental health that there is no evidence of selection processes in women, but there is in men. Myrskylä and Margolis [100] even assume that the effect of parenthood is underestimated rather than overestimated in cross-sectional studies, because they found stronger positive effects of parenthood on happiness in Germany in longitudinal analyses than in cross-sectional analyses. However, the results presented here do not clarify to what extent the bio-psycho-social situation has an influence on whether and at what age women and men have children (selectivity), or whether parenthood in different phases of life (also taking into account the number and age of children) strains or enhances health (causality) [4,101].

A further limitation is that the analysis presented here is based on data on persons in the common household; it can therefore only reflect social parenthood. Parents who do not live with their own children (e.g., after separation or divorce) cannot be identified. The same applies to parents whose children have already left their parents’ home. Particularly in the group of older respondents who do not live with a child in the household, it is therefore not possible to differentiate between childless women and men and parents whose children had already left the parental household. Therefore, it cannot be ruled out that some of the reported results are confounded by the fact that women and men who had already become parents around the age of 20 and who may be less healthy are responsible for the poorer health status of women and men without children in the household over the age of 40.

Furthermore, no information is available on the age of the respondents at the birth of the first child. It can be assumed that—in addition to the phase of life in which the women and men were when taking part at the survey—the age of the first child’s birth is also a decisive influencing factor, which is, moreover, associated with education and socio-economic situation [102]. Moreover, it could be important for health status whether parenthood was planned or childlessness was intended—especially since there are clear links with education [75]. Other important influencing factors can be assumed to be the quality of the parent-child relationship [22,103] or the parenting stage [104]. Simon and Caputo were able to show that parents whose children were 30 years and older reported better mental and physical health than parents of younger children [104]. 

There was also no analysis of whether the differences by age in the association between parental and health status varies with partner, employment or socioeconomic statuses, nor the region of residence (moderation effects). However, in a study by Margolis and Myrskylä [79] the moderating effect of income on age differences in the association between parenthood and happiness was small. It was only found in the group aged under-20 and in parents with three or more children. The authors have also shown that the age gradient in the association between parenthood and happiness is not moderated by partnership [79]. A moderating effect on age differences in the association between parental status and health could also be assumed with regard to employment status; however, there are no studies on this issue yet. For Germany in particular, it is also of interest for further analyses as to what extent there are differences between East and West Germany. Until the fall of communism in 1989, there were clear differences in reproductive biographies (e.g., a significantly younger age of mothers at the birth of their first child in East Germany). However, it can be assumed that this is more important with regard to the health status of older women and men, because since reunification there has been a convergence of reproductive biographies in East and West Germany [102]. In a longitudinal analysis using data from the German Socio-Economic Panel, Hank [105] was able to show that early motherhood in women aged 50 and over has been associated with poorer physical health in Western Germany, while late motherhood has been associated with lower psychological well-being in Eastern Germany. A comparison of Western and Eastern European countries has come to a similar finding. In both Western and Eastern Europe, early parenthood has been associated with poorer general health in later life, but the interrelationships have been much less pronounced in Eastern than in Western Europe [96]. 

Overall, a deeper understanding of the association between parenthood and health throughout the life course requires longitudinal analyses and a more precise distinction between fertility history and phases of living with children or social parenting. 

## 5. Conclusions

The present analysis shows that the association between living with children and the health and health behavior of women and men varied greatly with age. This was particularly true for women, where younger mothers were less healthy or behaved in less healthy ways and older mothers were healthier or had healthier behaviors than women of the same age without children. Among men, this pattern was similar only for health behavior and it was slightly less pronounced for self-rated general health and overweight.

Thus, living together with children is an important social determinant of health strongly associated with age which is relevant for both research and health promotion. 

With regard to health promotion for parents, a mixture of measures at the individual, the communal as well as the societal level seems promising—especially targeting young parental caregivers up to the age of 25. As Myrskylä and Margolis [100] can verify for Germany that changes in family policy (extension of parental leave benefits in 2007) were accompanied by a significant increase in the well-being of parents in the period around the birth of a child, measures at the societal level could play an important role in improving the health of parents [106]. Thus, Lee et al. [107] have shown that California’s paid family leave policy had positive impacts on several parental health outcomes such as self-rated health, overweight, psychological distress, and alcohol consumption. As there are large differences between countries in family leave policies, the effects on parents’ health would be an interesting topic for future studies [108,109]. It would also be of interest to analyze whether there are differences in the effects of the policy measures by age.

Nomaguchi and Milkie [3] emphasize the importance of parents’ welfare for the parents themselves as well as for child development and the overall health of society.

## Figures and Tables

**Figure 1 ijerph-17-03180-f001:**
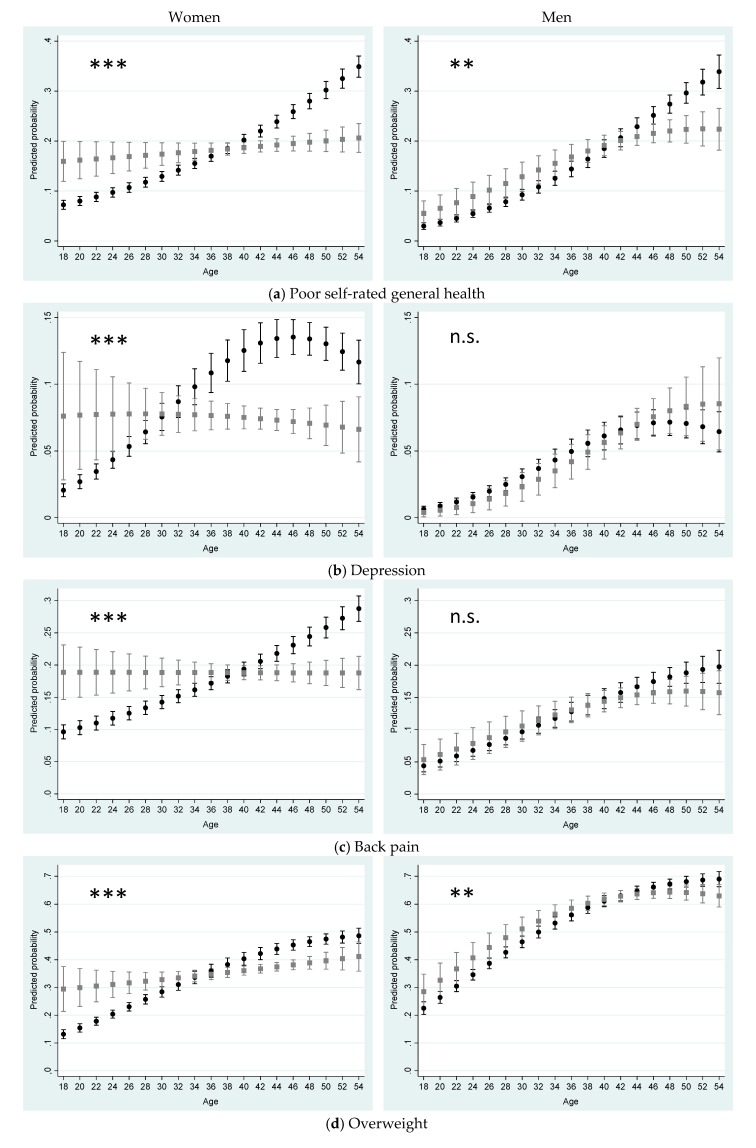
Health of women and men by parental status and age (in completed years) (predicted probabilities in %, 95% CI, adjusted for partner, socioeconomic and employment statuses, pre-school child in the household, region of residence). ■ Child in the household; ● No child in the household; Interaction between parental status and age: * *p* < 0.05; ** *p* < 0.01; *** *p* < 0.001; n.s.: not significant.

**Table 1 ijerph-17-03180-t001:** Description of the sample.

Variables	Women			Men		
	n Unweighted	% Weighted	Missing % Unweighted	n Unweighted	% Weighted	Missing % Unweighted
Total	21,379	100.0	0	17,717	100.0	0
**Outcome variables**						
*Poor self-rated general health*			0.1			0.1
Yes	3921	19.9		2663	17.5	
No	17,442	80.1		15,044	82.5	
*Depression*			0.3			0.1
Yes	1931	8.6		889	5.2	
No	19,382	91.4		16,804	94.8	
*Back pain*			0.1			0.1
Yes	3868	18.9		2115	13.3	
No	17,484	81.1		15,580	86.7	
*Overweight*			2.8			1.0
Yes	6875	35.0		9054	53.7	
No	13,898	65.0		8,486	46.3	
*Smoking*			<0.1			<0.1
Yes	6780	33.4		6613	40.0	
No	14,590	66.6		11,100	60.0	
*Sporting inactivity*			0.1			<0.1
Yes	5490	29.2		4451	29.0	
No	15,878	70.8		13,260	71.0	
**Predictor and control variables**						
*Age groups*			0			0
18–24 years	3221	15.9		3262	16.2	
25–34 years	4408	24.1		3650	24.0	
35–44 years	6771	27.1		5187	27.0	
45–54 years	6979	33.0		5618	32.8	
*Parental status*			0			0
Child(ren) in household	9731	55.3		6082	37.0	
No child in household	11,648	44.7		11,635	63.0	
*Pre-school child in household (0–6 years)*			0			0
Yes	4109	19.2		2688	15.2	
No	17,270	80.8		15,029	84.8	
*Partner status*			0.4			0.6
Partner	13,022	67.6		10,097	63.7	
No partner	8266	32.3		7521	36.3	
*Socioeconomic status*			0.2			0.2
High	6492	16.0		6120	17.0	
Middle	12,608	61.6		9680	59.2	
Low	2236	22.4		1889	23.8	
*Employment status*			0.5			0.4
Full-time employed	7919	34.2		13,581	76.4	
Part-time employed	8496	39.8		1692	9.1	
Not employed	4867	26.0		2379	14.5	
*Residential region*			0			0
West Germany	17,195	80.5		14,216	79.9	
East Germany	4184	19.5		3501	20.1	
*Survey year*			0			0
2009	7909	37.5		6058	34.2	
2010	8186	38.5		6269	34.6	
2012	5284	24.0		5390	31.2	

**Table 2 ijerph-17-03180-t002:** Health of women and men by parental status, stratified by age groups and sex (prevalence in %, 95% CI, *p*-values).

**Women**	**Child in Household**	**Total**			**18–24 Years**			**25–34 Years**			**35–44 Years**			**45–54 Years**		
		**%**	**95% CI**	***p***	**%**	**95% CI**	***p***	**%**	**95% CI**	***p***	**%**	**95% CI**	***p***	**%**	**95% CI**	***p***
Poor general health	No	21.3	20.4–22.2	***	11.6	10.5–12.9		13.5	11.8–15.3	*	21.6	19.3–24.1	**	31.4	29.7–33.1	***
	Yes	18.1	17.1–19.1		13.1	8.8–19.2		16.5	14.7–18.5		17.7	16.4–19.1		21.1	19.1–23.4	
Depression	No	9.7	9.1–10.4	***	4.3	3.6–5.2	*	8.9	7.6–10.3		11.7	9.9–13.6	***	13.1	12.0–14.3	***
	Yes	7.2	6.6–7.9		7.7	4.4–13.0		7.1	5.9–8.4		7.1	6.3–8.0		7.6	6.4-9.0	
Back pain	No	19.1	18.3–20.0		12.6	11.4–13.9	***	13.9	12.2–15.7	***	19.4	17.2–21.9		25.9	24.3–27.5	***
	Yes	18.6	17.7–19.6		27.1	20.5–34.8		18.7	16.8–20.6		18.1	16.8–19.4		19.2	17.2–21.3	
Overweight	No	33.4	32.4–34.5	***	15.7	14.3–17.2	***	26.7	24.5–29.0	***	35.9	33.1–38.8		47.6	45.8–49.4	***
	Yes	36.8	35.7–38.0		32.3	25.1–40.4		37.0	34.6–39.4		36.3	34.7–38.0		38.0	35.5–40.5	
Smoking	No	36.6	34.6–36.7	***	32.8	31.0–34.6	***	35.6	33.2–38.0		39.1	36.3–42.0	***	36.5	34.8–38.2	***
	Yes	30.6	29.5–31.8		49.0	40.9–57.1		33.7	31.4–36.1		30.7	29.2–32.3		25.1	23.0–27.4	
Sporting inactivity	No	25.9	24.9–26.9	***	14.4	13.1–15.9	***	21.2	19.1–23.4	***	31.1	28.4–34.0		34.2	32.5–35.9	***
	Yes	33.4	32.2–34.6		38.0	30.4–46.2		39.9	37.5–42.3		32.6	31.0–34.3		26.0	23.7–28.4	
**Men**	**Child in Household**	**Total**			**18–24 Years**			**25–34 Years**			**35–44 Years**			**45–54 Years**		
		**%**	**95% CI**	***p***	**%**	**95% CI**	***p***	**%**	**95% CI**	***p***	**%**	**95% CI**	***p***	**%**	**95% CI**	***p***
Poor general health	No	17.7	16.8–18.6		7.9	6.9–9.0		10.9	9.4–12.5	*	20.8	18.4–23.3	**	29.7	27.8–31.7	***
	Yes	17.1	15.9–18.4		8.8	3.0–22.8		14.1	11.3–17.4		15.8	14.2–17.7		20.6	18.3–23.0	
Depression	No	5.5	5.0–6.0		2.2	1.7–2.8		4.7	3.8–5.8	**	7.6	6.2–9.2	***	7.8	6.7–8.9	
	Yes	4.6	3.9–5.4		3.9	0.8–17.0		1.8	1.0–3.1		4.3	3.4–5.3		6.8	5.3–8.6	
Back pain	No	12.7	11.9–13.5	*	6.6	5.7–7.6	*	9.6	8.2–11.2		14.6	12.6–16.8		19.1	17.5–20.9	*
	Yes	14.3	13.2–15.4		16.4	7.7–31.7		11.3	9.2–13.8		14.2	12.8–15.9		15.9	13.9–18.1	
Overweight	No	48.8	47.6–50.0	***	25.0	23.3–26.7		42.0	39.7–44.3	***	58.5	55.9–61.2	*	68.3	66.4–70.2	
	Yes	62.2	60.7–63.6		29.7	17.5–45.6		54.8	50.9–58.7		62.8	60.7–64.8		65.9	63.4–68.3	
Smoking	No	41.4	40.3–42.6	***	37.3	35.5–39.1	***	44.8	42.5–47.2		45.6	42.9–48.4	***	39.8	37.8–41.9	***
	Yes	37.5	36.0–39.1		70.5	54.8–82.4		47.5	43.6–51.4		36.8	34.7–39.0		32.4	29.9–35.1	
Sporting inactivity	No	27.3	26.2–28.3	***	11.1	9.9–12.4	***	22.0	20.1–24.1	***	34.8	32.2–37.6		40.7	38.7–42.8	***
	Yes	32.0	30.5–33.5		39.4	25.2–55.6		33.4	29.8–37.2		33.1	31.0–35.3		29.6	27.1–32.2	

* *p* < 0.05; ** *p* < 0.01; *** *p* < 0.001.

**Table 3 ijerph-17-03180-t003:** Results of the logistic regressions with interaction of parental status and age for women (odds ratios, p values; age (metric) centered at the minimum of 18 years).

Women	Poor Self-Rated General Health		Depression		Back Pain		Overweight		Smoking		Sporting Inactivity	
	OR	*p*	OR	*p*	OR	*p*	OR	*p*	OR	*p*	OR	*p*
**Parental status: Child in household**												
No child	Ref.		Ref.		Ref.		Ref.		Ref.		Ref.	
Child	2.43	<0.001	3.92	<0.001	2.19	<0.001	2.75	<0.001	6.15	<0.001	3.39	<0.001
**Age**	1.06	<0.001	1.16	<0.001	1.04	<0.001	1.10	<0.001	1.08	<0.001	1.13	<0.001
**Age#Age**			0.997	<0.001			0.999	<0.001	0.998	<0.001	0.998	<0.001
**Child#Age**												
No child#Age	Ref.		Ref.		Ref.		Ref.		Ref.		Ref.	
Child#Age	0.96	<0.001	0.87	<0.001	0.96	<0.001	0.92	<0.001	0.88	<0.001	0.91	<0.001
**Child#Age#Age**												
No child#Age#Age			Ref.				Ref.		Ref.		Ref.	
Child#Age#Age			1.002	0.005			1.001	0.013	1.002	<0.001	1.001	0.016
**Pre-school child in household**												
No	Ref.		Ref.		Ref.		Ref.		Ref.		Ref.	
Yes	0.61	<0.001	0.68	0.004	0.91	0.265	1.02	0.770	0.55	<0.001	1.42	<0.001
**Partner status**												
Partner	Ref.		Ref.		Ref.		Ref.		Ref.		Ref.	
No partner	1.15	0.007	1.85	<0.001	1.13	0.023	0.79	<0.001	1.45	<0.001	0.92	0.076
**Socioeconomic status**												
High	Ref.		Ref.		Ref.		Ref.		Ref.		Ref.	
Middle	1.78	<0.001	1.21	0.007	1.55	<0.001	1.75	<0.001	1.65	<0.001	2.13	<0.001
Low	2.92	<0.001	1.39	0.002	2.17	<0.001	2.51	<0.001	2.68	<0.001	4.56	<0.001
**Employment status**												
Full-time	Ref.		Ref.		Ref.		Ref.		Ref.		Ref.	
Part-time	1.06	0.278	1.49	<0.001	0.97	0.590	0.87	0.002	0.83	<0.001	0.78	<0.001
Not employed	2.07	<0.001	2.66	<0.001	1.35	<0.001	1.05	0.404	0.78	<0.001	1.10	0.071
**Residential region**												
West Germany	Ref.		Ref.		Ref.		Ref.		Ref.		Ref.	
East Germany	0.99	0.868	0.97	0.706	1.04	0.465	1.06	0.208	1.01	0.836	1.10	0.053

**Table 4 ijerph-17-03180-t004:** Results of the logistic regressions with interaction of parental status and age for men (odds ratios, *p* values; age (metric) centered at the minimum of 18 years).

Men	Poor Self-Rated General Health		Depression		Back Pain		Overweight		Smoking		Sporting Inactivity	
	OR	*p*	OR	*p*	OR	*p*	OR	*p*	OR	*p*	OR	*p*
**Parental status: Child in household**												
No child	Ref.		Ref.		Ref.		Ref.		Ref.		Ref.	
Child	1.53	0.022	0.70	0.267	1.13	0.520	1.27	0.055	3.98	<0.001	4.15	<0.001
**Age**	1.12	<0.001	1.19	<0.001	1.09	<0.001	1.11	<0.001	1.09	<0.001	1.17	<0.001
**Age#Age**	0.999	0.007	0.997	<0.001	0.999	0.005	0.999	<0.001	0.998	<0.001	0.997	<0.001
**Child#Age**												
No child#Age									Ref.		Ref.	
Child#Age									0.886	<0.001	0.898	0.001
**Child#Age#Age**												
No child#Age#Age	Ref.		Ref.		Ref.		Ref.		Ref.		Ref.	
Child#Age#Age	0.999	<0.001	1.001	0.175	0.9997	0.164	0.9996	0.006	1.002	0.002	1.002	0.038
**Pre-school child in household**												
No	Ref.		Ref.		Ref.		Ref.		Ref.		Ref.	
Yes	0.75	0.010	0.59	0.007	0.90	0.356	0.93	0.332	0.81	0.015	1.04	0.651
**Partner status**												
Partner	Ref.		Ref.		Ref.		Ref.		Ref.		Ref.	
No partner	1.14	0.076	1.57	<0.001	0.84	0.028	0.82	<0.001	1.20	0.001	1.17	0.009
**Socioeconomic status**												
High	Ref.		Ref.		Ref.		Ref.		Ref.		Ref.	
Middle	2.48	<0.001	1.57	<0.001	1.77	<0.001	1.46	<0.001	1.77	<0.001	2.53	<0.001
Low	3.93	<0.001	2.04	<0.001	2.43	<0.001	2.00	<0.001	2.55	<0.001	4.53	<0.001
**Employment status**												
Full-time	Ref.		Ref.		Ref.		Ref.		Ref.		Ref.	
Part-time	1.61	<0.001	2.27	<0.001	1.28	0.024	0.68	<0.001	0.93	0.326	0.90	0.230
Not employed	3.47	<0.001	3.99	<0.001	2.04	<0.001	0.78	<0.001	0.87	0.035	0.98	0.744
**Residential region**												
West Germany	Ref.		Ref.		Ref.		Ref.		Ref.		Ref.	
East Germany	0.88	0.048	0.75	0.009	0.90	0.134	0.92	0.083	1.11	0.034	1.16	0.007

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
