# Peer review of "Association between Living with Children and the Health and Health Behavior of Women and Men. Are There Differences by Age? Results of the “German Health Update” (GEDA) Study"

_ijerph, 2020, doi:10.3390/ijerph17093180_

Round 1
Reviewer 1 Report
Dear authors,
it was my pleasure to review your comprehensive manuscript. The considered topic is of great interest nowadays due to the low birth rate in many European countries. The reviewed study is socially important as a possible motivation for having children.
I especially like the exhaustive state-of-the-art section with 95 sources. It gives a solid ground for the hypothesis and subject of the study. My only recommendation here is to refresh the list of references with some recent publications (less than 5 y.o.).
The design of the research is simple but elegant. The chosen methods are relevant to the study. My only concern is the relatively high age of sample data (2009-2012). It is clear, that collecting such amount of data is a very lengthy process, keeping in mind the sample size and processing cost. However, I recommend clarifying this moment in the paper.
Nevertheless, my impression is great and I can recommend the paper for publication after only minor revisions.
Reviewer 2 Report
Association between living with children and the health and health behavior of women and men. Are there differences by age? Results of the “German Health Update” (GEDA) study (ijerph-781190)
Main message of the article
This article investigates the health status and behavior of German adults and compares the likelihood of suffering of physical and health pathologies, as well as the probability of engaging in unhealthy behaviors, across men and women living with and without children.
General Judgment Comments
Overall, the article is well written and scientifically sound. The high number of participants and the usage of a known dataset give strength to the quality of work here presented.
Suggestion: Minor revision
The article is clear and well written. Despite that, some minor edits are suggested before the article can be accepted for publication.
Major Issues
- On-Line 127, the authors indicated that they restricted the sample to participants aged 18 to 54. What is the rationale behind this selection? Given that the sample size decreased by almost 50% following this decision, additional information should be provided in order to explain the authors' reasoning in this step.
- Similarly, on line 151, the authors’ indicated that the results are stratified by age groups, however, no clear indication about how they decided to divide the groups is given. Is this an arbitrary decision or is it based on the percentages of individuals per group? The authors should consider adding a brief explanation here as well.
- Figure 1, the asterisks are covered by the image (for example in Smoking). Not sure whether this is just a rendering problem, but it’s currently impossible to identify the significant comparisons by the visual aids on the images.
Minor Issues
- Given what the authors say about the way they defined parenting accordingly to the data available to them, the word caregiver seems more appropriate than the parent within the context, but some freedom can here be left to the authors since they explain the way they defined the variable.
- Stylistically, the last phrase of the manuscript sounds a bit rude and seems to point out a general lack of attention by the researchers within the field. I suggest replacing the phrase with something like “Future studies should address …”.
- As data have been aggregated from different rounds of the same dataset, it would have been interesting to see how the number of participants per year concurring in the final number of participants (for example on Line 129).
- For the dichotomous variables presented in the first section of Table 1, it took me a couple of reads to understand what the “% weighted” stands for. While it’s clear for the below sections, additional information about what the percentage refers to will enhance the readability of the table.
Final comments
The manuscript is overall of very good quality, well written and scientifically sound. The authors did a great job of aggregating data and analyze their dataset. The results are of general interests and there is a clear and well-defined pattern for future research on the topic, such as in comparing data from East and West Germany. However, before accepting the paper for publication, I recommend the editor to ask for the correction of minor errors and unclear points present within the text.
